# Flexible mean field variational inference using mixtures of non-overlapping exponential families

**Jeffrey P. Spence**
Stanford University
Stanford, CA 94305
`jspence@stanford.edu`

## Abstract

Sparse models are desirable for many applications across diverse domains as they can perform automatic variable selection, aid interpretability, and provide regularization. When fitting sparse models in a Bayesian framework, however, analytically obtaining a posterior distribution over the parameters of interest is intractable for all but the simplest cases. As a result practitioners must rely on either sampling algorithms such as Markov chain Monte Carlo or variational methods to obtain an approximate posterior. Mean field variational inference is a particularly simple and popular framework that is often amenable to analytically deriving closed-form parameter updates. When all distributions in the model are members of exponential families and are conditionally conjugate, optimization schemes can often be derived by hand. Yet, I show that using standard mean field variational inference can fail to produce sensible results for models with sparsity-inducing priors, such as the spike-and-slab. Fortunately, such pathological behavior can be remedied as I show that mixtures of exponential family distributions with non-overlapping support form an exponential family. In particular, any mixture of an exponential family of diffuse distributions and a point mass at zero to model sparsity forms an exponential family. Furthermore, specific choices of these distributions maintain conditional conjugacy. I use two applications to motivate these results: one from statistical genetics that has connections to generalized least squares with a spike-and-slab prior on the regression coefficients; and sparse probabilistic principal component analysis. The theoretical results presented here are broadly applicable beyond these two examples.

## 1 Introduction

Bayesian graphical models [36] are widely used across vast swaths of engineering and science. Graphical models succinctly summarize modeling assumptions, and their posterior distributions naturally quantify uncertainty and produce optimal point estimators to minimize downstream risk. Efficient, exact posterior inference algorithms exist for special cases such as the sum-product algorithm [45] for discrete random variables in models with low tree-width [61] or particular Gaussian graphical models [30]. Outside of these special cases, however, alternative techniques must be used to approximate the posterior. Markov chain Monte Carlo (MCMC), especially Gibbs sampling, can sample from the posterior asymptotically [20, 23] but such techniques are difficult to scale to models with many observations, can have issues with mixing [43], and their convergence can be hard to assess [56].

Variational inference (VI) [29] avoids sampling and instead fits an approximate posterior via optimizing a loss function that acts as a proxy for some measure of divergence between the approximate and true posteriors. Usually this divergence is the "reverse" Kullback-Leibler (KL) divergence between

the approximate posterior $Q$, and the true posterior $P$: $\mathrm{KL}(Q||P)$. Minimizing this reverse KL is equivalent to maximizing the so-called **e**vidence **l**ower **bo**und (ELBo) which depends only on the prior, the likelihood, and the approximate posterior. Importantly, the unknown true posterior does not appear in the ELBo. In many models, by choosing a particular space of approximate posteriors over which to optimize (the "variational family"), maximizing the ELBo is a tractable albeit non-convex optimization problem. In practice, such optimization problems are solved using coordinate ascent, gradient ascent, or natural gradient ascent [2]. These optimization methods and stochastic variants thereof can scale to massive models and datasets [25].

This formulation introduces a key tension in VI. On one hand, simpler variational families can be more computationally tractable to optimize. On the other hand, a large approximation error will be incurred if the true posterior does not lie within the variational family. VI is particularly tractable for fully conjugate models with all distributions belonging to exponential families, with a variational family that assumes all variables are independent. This approach, called *mean field VI*, has a number of desirable properties including computationally efficient, often analytic coordinate-wise updates that are equivalent to natural gradient steps [6]. Many useful models–including mixture models– are not fully conjugate, but can be made conjugate by introducing additional auxiliary variables. Unfortunately, introducing more variables and enforcing their independence in the variational family results in larger approximation gaps [54].

Many approaches have been developed to extend VI beyond the mean field approximation. Specific models, such as coupled hidden Markov models, are amenable to more expressive variational families because subsets of the variables form tractable models – an approach called structured VI [51]. For non-conjugate models or models with variational families that are otherwise not amenable to mean field VI, it may be possible to compute gradients numerically [35] or obtain an unbiased noisy estimate of the gradient via the log gradient trick (also called the REINFORCE estimator) [49] or in certain special cases the reparameterization trick [34]. Because such approaches are broadly applicable they have come to be part of the toolkit of "black box VI" [48], which seeks to automatically derive gradient estimators for very general models and variational families. Stochastic gradient-based optimization has also been combined with structured VI [24]. Additionally, approximations to gradients tailored to specific distributions have been developed [27]. Much recent work has been devoted to combining deep learning with Bayesian graphical models, whether purely for speeding up inference, or by using neural networks as flexible models [5, 28, 60]. Overall, these general approaches are impressive in scope, but the gradient estimators can be noisy and hence optimizing the ELBo can require small step sizes and corresponding long compute times [50].

Here, I take an alternative approach, expanding the utility of mean field VI by showing that more flexible variational families can be constructed by combining component exponential families while maintaining desirable conjugacy properties and still forming an exponential family. This construction is particularly useful in models involving sparsity. The approach I present maintains the analytic simplicity of mean field VI but avoids the need to introduce auxiliary variables to obtain conjugacy.

I begin with a motivating example in Section 2, develop the main theoretical results in Section 3, and show their utility on two examples in Section 4. Implementation details, proofs, and additional theoretical results are presented in the Appendix.

## 2 A Motivating Example

To better understand the genetics of disease and other complex traits, it is now routine to collect genetic information and measure a trait of interest in many individuals. Initially, such studies hoped to find a small number of genomic locations associated with the trait to learn about the underlying biology or to find suitable drug targets. As more of these genome-wide association studies (GWAS) were performed, however, it became increasingly clear that enormous numbers of locations throughout the genome contribute to most traits, including many diseases [8]. While this makes it difficult to prioritize a small number of candidate genes for biological followup, one can still use GWAS data to predict a trait from an individual's genotypes, an approach referred to as a polygenic score (PGS) [14]. For many diseases, PGSs are sufficiently accurate to be clinically relevant for stratifying patients by disease risk [32].

Typically PGSs are constructed by estimating effect sizes for each position in the genome and assuming that positions contribute to the trait additively. This assumption is justified by evolutionary

arguments [52] and practical concerns about overfitting due to the small number of individuals in GWAS (tens to hundreds of thousands) relative to the number of genotyped positions (millions). More concretely, the predicted value of the trait $\hat{Y}_i$ in individual $i$ is

$$\hat{Y}_i := \sum_{j=1}^{P} G_{ij}\beta_j$$

where $G_{ij}$ is the genotype of individual $i$ at position $j$ and $\beta_j$ is the effect size at position $j$.

Estimating the effect sizes, $\beta_j$, is complicated by the fact that to protect participant privacy GWAS typically only release marginal effect size estimates $\hat{\beta}_j$ obtained by separately regressing the value of the trait against the genotypes at each position. These marginal estimates thus ignore the correlation in individuals' genotypes at different positions in the genome. Fortunately, these correlations are stable across sets of individuals from the same population and can be estimated from publicly available genotype data (e.g., [12]) even if that data does not contain measurements of the trait of interest.

An early approach to dealing with genotypic correlations was LDpred [57] which uses MCMC to fit the following model with a spike-and-slab prior on the effect sizes:

$$\beta_j \overset{\text{i.i.d.}}{\sim} p_0\delta_0 + (1-p_0)\mathcal{N}(0, \sigma_g^2)$$
$$\hat{\beta}|\beta \sim \mathcal{N}(\mathbf{X}\beta, \sigma_e^2\mathbf{X}),$$

where $\mathbf{X} \in \mathbb{R}^{P \times P}$ is the matrix of correlations between genotypes at different positions in the genome, $\sigma_e^2$ is the variance of the estimated effect sizes from the GWAS and $\sigma_g^2$ is the prior variance of the non-zero effect sizes.

While this model was motivated by PGSs, it is also a special case of Bayesian generalized least squares with a spike-and-slab prior on the regression coefficients.

Note that if $\mathbf{X}$ is diagonal, then the likelihood is separable and the true posterior can, in fact, be determined analytically. Unfortunately, in real data genotypes are highly correlated (tightly linked) across positions and so $\mathbf{X}$ is far from diagonal and as a result integrating over the mixture components for each $\beta_j$ must be done simultaneously, resulting in a runtime that is exponential in the number of genotyped positions. To obtain a computationally tractable approximation to the true posterior, LDpred and a number of extensions that incorporate more flexible priors (e.g., [19, 37]) use MCMC. As noted above, however, MCMC has a number of drawbacks and so there may be advantages to deriving and implementing a VI scheme.

A natural mean field VI approach to approximating the posterior would be to split the mixture distribution using an auxiliary random variable as follows

$$Z_j \overset{\text{i.i.d.}}{\sim} \text{Bernoulli}(1-p_0)$$
$$\beta_j|Z_j \sim \mathcal{N}(0, \sigma_{Z_j}^2) \tag{1}$$
$$\hat{\beta}|\beta \sim \mathcal{N}(\mathbf{X}\beta, \sigma_e^2\mathbf{X}),$$

with $\sigma_0^2 = 0$ and $\sigma_1^2 = \sigma_g^2$ (treating a Gaussian with zero variance as a point mass for notational convenience). One can then use categorical variational distributions $q_{Z_j}$ for the $Z$s and Gaussian variational distributions $q_{\beta_j}$ for the $\beta$s. Unfortunately, this approach immediately encounters an issue: when calculating the ELBo, $\text{KL}(q_{\beta_j}||\mathcal{N}(0,0))$ is undefined unless $q_{\beta_j}$ is a point mass at zero because otherwise $q_{\beta_j}$ is not absolutely continuous with respect to the point mass at zero. This may seem to be merely a technical issue that vanishes if $\sigma_0^2$ is taken to be some tiny value so that $q_{\beta_j}$ is absolutely continuous with respect to $\mathcal{N}(0, \sigma_0^2)$ and hence has well-defined KL, while for practical purposes $\mathcal{N}(0, \sigma_0^2)$ acts like a point mass at zero. Yet, this superficial fix is not enough: the mean field assumption requires $Z_j$ and $\beta_j$ to be independent under the variational approximation to the posterior, which cannot capture the phenomenon that $Z_j = 0$ forces $\beta_j$ to be close to zero while $Z_j = 1$ allows $\beta_j$ to be far from zero. Further intuition is presented in Appendix A, where I analyze the case with only a single position (i.e., $P = 1$) in detail.

Fortunately, this problem can be solved by noting that spike-and-slab distributions like $p_0\delta_0 + (1-p_0)\mathcal{N}(0, \sigma_g^2)$ surprisingly still form an exponential family and are conjugate to the likelihood. In this

example, by using a spike-and-slab distribution for the variational distributions $q_{\beta_j}$, there is no need to use auxiliary variables to obtain analytical updates for fitting the approximate posterior. A similar approach has been considered for a specific model in a different context [10], but in the next section, I show why the analysis works for this approach and also show that it is more broadly applicable. In Section 4, I return to this motivating example to show that the naive mean field VI does indeed break down and that by using sparse conjugate priors an accurate approximation to the posterior may be obtained.

## 3    More flexible exponential families

In this section I begin with a simple observation. While it is usually true that mixtures of distributions from an exponential family no longer form an exponential family, that is not the case for mixtures of distributions with distinct support. Mixtures of exponential family members with non-overlapping support always form an exponential family.

**Theorem 1.** *Let $F_1, \ldots, F_K$ be exponential families of distributions, where the distributions within each family have supports $\mathcal{S}_1, \ldots, \mathcal{S}_K$ such that $\mathcal{S}_i \cap \mathcal{S}_j = \varnothing$ for all $i \neq j$. Further, let $\eta_1, \ldots, \eta_K$ be the natural parameters of the exponential families, $T_1, \ldots, T_K$ be the sufficient statistics, $A_1, \ldots, A_K$ be the log-partitions, and $H_1, \ldots, H_K$ be the base measures. Then the family of mixture distributions*

$$F_{mix} = \left\{ \sum_{k=1}^{K} \pi_k f_k : \sum_{k=1}^{K} \pi_k = 1, \pi_i \geq 0, f_i \in F_i, \forall i \right\}$$

*is an exponential family with natural parameters*

$$\eta_{mix} = \big(\eta_1, \ldots, \eta_K, \log \pi_1 - A_1(\eta_1) - \log \pi_K + A_K(\eta_K), \ldots,$$
$$\log \pi_{K-1} - A_{K-1}(\eta_{k-1}) - \log \pi_K + A_K(\eta_K)\big),$$

*corresponding sufficient statistics*

$$T_{mix}(x) = \left( \mathbb{I}\{x \in \mathcal{S}_1\} T_1(x), \ldots, \mathbb{I}\{x \in \mathcal{S}_K\} T_K(x), \mathbb{I}\{x \in \mathcal{S}_1\}, \ldots, \mathbb{I}\{x \in \mathcal{S}_{K-1}\} \right),$$

*log-partition*

$$A_{mix}(\eta_{mix}) = A_K(\eta_K) - \log \pi_K,$$

*and base-measure*

$$\frac{dH_{mix}}{dH}(x) = \prod_{i=1}^{K} \left( \frac{dH_i}{dH}(x) \right)^{\mathbb{I}\{x \in \mathcal{S}_i\}}$$

*where $H$ is a measure such that $H_i$ is absolutely continuous with respect to $H$ for all $i$ and $dH_i/dH$ is the usual Radon-Nikodym derivative and $0^0$ is taken to be 1.*

Note that Theorem 1 has an apparent asymmetry with respect to component $K$. This arises because the mixture weights $\pi_1, \ldots, \pi_K$ are constrained to sum to one, and hence $\pi_K$ is completely determined by the other mixture weights. It would be possible to have a "symmetric" version of Theorem 1 but the constraint on the mixture weights would mean that the natural parameters would live on a strictly lower dimensional space. Such exponential families are called *curved exponential families*, and many of the desirable properties of exponential families do not hold for curved exponential families.

While the restriction in Theorem 1 to exponential families with non-overlapping support may seem particularly limiting, it is important to note that a new exponential family can be formed by restricting all of the distributions within any exponential family to lie within a fixed subset of their usual domain and renormalizing. In particular, one could divide the original full domain into non-overlapping subsets and form mixtures of exponential families restricted to these subsets. Another important set of exponential families are uniform distributions with fixed support. While each of these exponential families only contains a single distribution, by combining mixtures of these uniform distributions with non-overlapping support, Theorem 1 shows that piece-wise constant densities with fixed break points form an exponential family. In this paper I focus primarily on the case of mixtures of a continuous exponential family member with one or more point masses. By Theorem 1, mixtures of point masses at distinct, fixed locations form an exponential family, and diffuse distributions can be trivially restricted to falling outside of this set of measure zero, which makes clear that spike-and-slab

distributions where the slab is a non-atomic distribution from an exponential family always form an exponential family.

While these substantially more flexible exponential families may be of independent interest, they are of little use in mean field VI unless they are conjugate to widely used likelihood models. The following theorem provides a recipe to take a conjugate exponential family model and create a model with a more flexible prior using Theorem 1 while maintaining conjugacy.

**Theorem 2.** *Let $F_{prior}(x)$ be an exponential family of prior distributions with sufficient statistics $T_{prior}(x)$ with the distributions supported on $\mathcal{S}_{prior}$ that are conjugate to an exponential family of distributions $F_{Y|X}(y|x)$. For exponential families $F_1, \ldots, F_K$ with supports $\mathcal{S}_1, \ldots, \mathcal{S}_K$ such that $\mathcal{S}_i \cap \mathcal{S}_j = \varnothing$ for all $i \neq j$, if there exists a matrix $M_i$ and vector $v_i$ such that $T_{prior}(x) = M_i T_i(x) + v_i$ for all $x \in \mathcal{S}_i$ for all $i$, then $F_{mix}$ as defined in Theorem 1 is also conjugate to $F_{Y|X}$.*

Intuitively, Theorem 2 says that for a given conjugate prior distribution, say $\mathcal{P}$, we can create a more flexible prior that maintains conjugacy by combining non-overlapping component distributions so long as those component distributions have the same sufficient statistics as $\mathcal{P}$ on their domain up to an affine transformation. An alternative way to state the theorem is that if each component exponential family is conjugate to a distribution and distributions in different component families have non-overlapping support, then the mixture distributions form a conjugate exponential family.

Note that the point mass distribution at a fixed point is a member of an exponential family with a sole member, and hence it has no sufficient statistics. Furthermore, any function is constant on the support of a point mass. This provides an immediate corollary that mixtures of any continuous distribution with a finite number of point-masses are conjugate to the same distributions as the original continuous distribution. Another example is mixtures of degenerate distributions where only a subset of parameters are fixed. For instance, consider the mean vector, $\mu$, of a multivariate Gaussian. One could put a prior on $\mu$ that is a mixture of the usual conjugate multivariate Gaussian along with degenerate distributions like a multivariate Gaussian with the condition that $\mu_i = 0$ or that $\mu_i = \mu_j$ for dimensions $i$ and $j$. In fact, any degenerate multivariate Gaussian distribution defined by affine constraints $\mathbf{M}\mu = \mathbf{v}$ for some matrix $\mathbf{M}$ and vector $\mathbf{v}$ could be included in this mixture.

Theorem 2 makes it extremely easy to add sparsity or partial sparsity to any conjugate model. An application using non-degenerate distributions is constructing asymmetric priors from symmetric ones. For example, we can construct an asymmetric prior by taking a mixture of a copy of the original prior restricted to the negative reals and a copy restricted to non-negative reals. These mixtures form an exponential family because the copies have the same sufficient statistics as the original prior but have non-overlapping domains.

## 4 Numerical results

To show the applicability of the theoretical results presented in Section 3, I test sparse VI schemes using Theorems 1 and 2–which I refer to as the non-overlapping mixtures trick–on two models and compare these schemes to non-sparse and naive VI approximations showing the superior performance of treating sparsity exactly. The first model is the polygenic score prediction model discussed in Section 2 and the second is a spike-and-slab prior for sparse probabilistic principal component analysis (PCA) [21]. Python implementations of the simulations, fitting procedures, and plotting are available at `https://github.com/jeffspence/non_overlapping_mixtures`. All mean field results were obtained on a Late 2013 MacBook Pro, and fitting the mean field VI schemes took less than five seconds per dataset for the polygenic score model and less than two minutes per dataset for sparse PCA.

### 4.1 Polygenic Scores

While the LDpred model was originally fit using Gibbs sampling [57], it may be desirable to fit the model using VI for computational reasons. I simulated data under the LDpred model to compare two VI schemes. The first VI scheme is obtained by using the non-overlapping mixtures trick. The second is the naive VI scheme of introducing auxiliary variables to split the spike-and-slab prior into a mixture, and then approximating that mixture as a mixture of Gaussians. The details of the VI schemes are presented in Appendix D. I simulated a 1000 dimensional vector $\hat{\beta}$ from the LDpred model with $p_0 = 0.99$ so that on average about 10 sites had non-zero effects, a level of sparsity

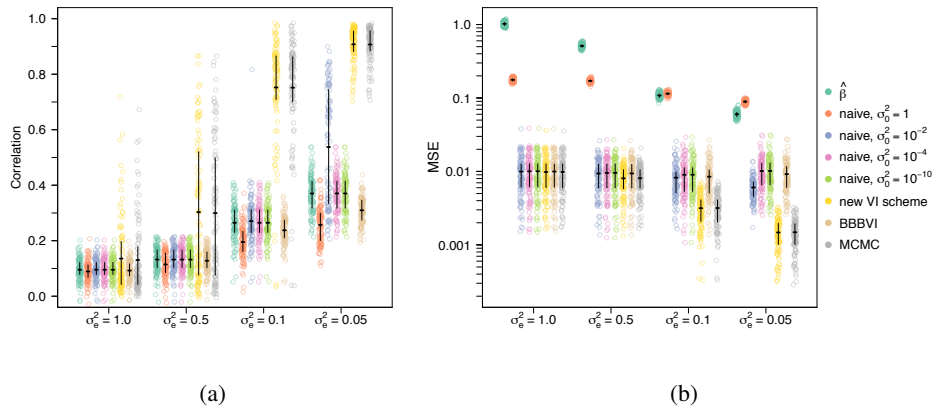

(a)　　　　　　　　　　　　　　　　　　(b)

Figure 1: **Comparison of VI schemes for the LDpred model**
The non-overlapping mixtures trick (new VI scheme) is compared to the naive scheme of introducing an auxiliary variable and approximating the spike-and-slab prior by a mixture of two gaussians centered at zero, the less dispersed of which has variance $\sigma_0^2$. As a baseline the VI schemes are compared against what is sometimes used in practice – the raw observations, $\hat{\beta}$ as well as boosting black box VI [38] and adaptive random walk MCMC. The parameter $\sigma_e^2$ controls the amount of noise, so $\sigma_e^2 = 0.05$ corresponds to a 20 times higher signal-to-noise ration than $\sigma_e^2 = 1.0$. Plot (a) compares the correlation between the estimates (posterior mean for the VI schemes, $\hat{\beta}$ for the baseline) and the true simulated values of $\beta$. Plot (b) compares the mean squared error (MSE). Point clouds are individual simulations, horizontal lines are means across simulations, and whiskers are interquartile ranges. See the main text for simulation details.

generally consistent with realistic human data [53]. For each simulation I drew $\mathbf{X}$ by simulating from a Wishart distribution with an identity scale matrix and 1000 degrees of freedom and then dividing the draw from the Wishart distribution by 1000. I set $\sigma_1^2$ to be 1 and then varied $\sigma_e^2$ from 0.05 to 1.0. For each of value of $\sigma_e^2$, I simulated 100 replicate datasets and tested the VI schemes as well as a baseline that is used in statistical genetics of using the raw values of $\hat{\beta}$ as estimates of $\beta$. I also tested adaptive random walk MCMC as implemented in NIMBLE [15] which was run for 1000 iterations, and boosting black box VI (BBBVI) [38] as implemented in `pyro` [4], which sequentially fits a mixture distribution as an approximate posterior. For BBBVI, I used the equivalent formulation $\hat{\beta}|\beta, Z \sim \mathcal{N}(\mathbf{X}(\beta * Z), \sigma^2 \mathbf{X})$, where the $\beta_i$ are independent Gaussians, and the $Z_i$ are independent Bernoullis, with $*$ being the component-wise product. For the component distributions of the variational family, I used independent Gaussians for the $\beta_i$ and independent Bernoullis for the $Z_i$. I used 2 particles to stochastically estimate gradients, 50 components in the mixture distribution, and 2000 gradient steps per mixture component. The BBBVI objective functions were optimized using the `adam` optimizer [33] with learning rate $10^{-3}$ and default parameters otherwise. I evaluated the methods using the mean squared error (MSE) and correlation between the estimated and true values of $\beta$, using the variational posterior mean as the estimator for the Bayesian approaches (Figure 1).

For the naive VI scheme, there is an additional parameter $\sigma_0^2$. When $\sigma_0^2 = 1 = \sigma_1^2$, the method is equivalent to performing mean field VI on the non-sparse model where the prior on $\beta$ is simply a single Gaussian, which is a Bayesian version of ridge regression. In addition to $\sigma_0^2 = 1$, I used $\sigma_0^2 \in \{10^{-2}, 10^{-4}, 10^{-10}\}$. The results for $\sigma_0^2 = 10^{-4}$ and $\sigma_0^2 = 10^{-10}$ are indistinguishable. All of the VI schemes (both the new scheme and the naive scheme with any value of $\sigma_0^2$) outperform the baseline of just using $\hat{\beta}$ except for in the extremely high signal-to-noise regime, where the naive model with $\sigma_0^2 = 1$ over-shrinks and cannot take advantage of sparsity. BBBVI performed comparably to the naive VI schemes, but BBBVI required hours to run compared to seconds for the mean field schemes. Meanwhile, by these metrics the non-overlapping mixtures trick performed indistinguishably from MCMC, but again took seconds per run compared to approximately 12 hours per run for MCMC. I also considered the maximum likelihood estimator (MLE) $\beta_{\text{MLE}} = \mathbf{X}^{-1}\hat{\beta}$ but $\mathbf{X}$ is terribly ill-conditioned resulting in high variance. Using the MLE almost always resulted in

correlations around zero (maximum correlation across all simulations was 0.13, mean correlation was 0.005) and extremely large MSE (minimum MSE across all simulations was 4.19, mean MSE was $\approx 335000$ – about six orders of magnitude higher than any other method). None of the naive schemes provide a substantial improvement over the baseline in terms of correlation. In terms of MSE, the naive schemes provide some improvement if $\sigma_0^2$ is tuned properly, but the non-overlapping mixture trick outperforms all of the schemes across signal-to-noise regimes. Exactly accounting for sparsity when the true signal is sparse substantially improves performance.

## 4.2 Sparse Probabilistic PCA

PCA [46, 26] is a widely-used exploratory data analysis method that projects high dimensional data into a low dimensional subspace define by orthogonal axes that explain a maximal amount of empirical variance. These axes are defined by "loadings"–weightings of the dimensions that compose a datapoint. Dimensions with high loadings in the first few PCs are deemed "important" for differentiating the data points. Unfortunately, the loadings are dense, making them difficult to interpret especially for high dimensional data.

To aid interpretability, and to leverage that in many datasets only a few variables are expected to contribute meaningfully to variation between data points, formulations of sparse PCA were developed to encourage sparsity in the loadings, usually by means of $\ell_1$ regularization [62].

In a parallel line of work, a Bayesian interpretation of PCA, probabilistic PCA, was developed by showing that classical PCA can be derived as the limiting maximum *a posteriori* estimate of a particular generative model up to possible scaling and rotation [55]. The probabilistic formulation more naturally extends to non-Gaussian noise models [11], allows principled methods for choosing the number of principal components [41], gracefully handles missing data [55], and enables speedups for structured datasets [1].

The probabilistic PCA model is

$$Z_1, \ldots, Z_N \sim \mathcal{N}(0, \mathbf{I}_K)$$
$$X_n | Z_n \sim \mathcal{N}(\mathbf{W} Z_n, \sigma_e^2 \mathbf{I}_P)$$

where $K$ is the number of PCs desired, $\mathbf{W} \in \mathbb{R}^{P \times K}$ is the matrix of loadings, and $Z_n$ is the PC score (i.e., projection onto the first $K$ PCs) of the $n^{\text{th}}$ datapoint.

These two lines of work were then brought together in sparse probabilistic PCA [21], which encourages sparse loadings by putting a Laplace prior on each loading, $\mathbf{W}_{pk}$. The Laplace prior is not conjugate to the Gaussian noise model, however, but the Laplace distribution is a scale mixture of Gaussians allowing for a hierarchical decomposition, making this formulation amenable to mean field VI.

There are a number of conceptually displeasing aspects to this formulation of sparse probabilistic PCA. First, while it is true that the maximum *a posteriori* estimate of the loadings is sparse under a Laplace prior, the generative model is not sparse: because the Laplace distribution is diffuse, the loadings are non-zero almost surely. Second, the LDpred model discussed in Section 2 is a discrete scale mixture of Gaussians with two components. I showed numerically in Section 4.1 and theoretically in Appendix A that mean field VI breaks down in such a setting suggesting that performing mean field VI on a scale mixture of Gaussians may be problematic especially in sparsity-inducing cases.

Motivated by these considerations I consider a spike-and-slab prior on the loadings:

$$\mathbf{W}_{pk} \sim p_0 \delta_0 + (1 - p_0) \mathcal{N}(0, \sigma_1^2)$$
$$Z_n \sim \mathcal{N}(0, \mathbf{I}_K)$$
$$X_n | Z_n, \mathbf{W} \sim \mathcal{N}(\mathbf{W} Z_n, \sigma_e^2 \mathbf{I}_P)$$

Note that [21] considered a fully Bayesian model, which here would correspond to putting uninformative conjugate priors on $p_0$, $\sigma_e^2$, and $\sigma_1^2$. For ease of exposition, I consider those to be fixed hyperparameters, but future work could explore putting priors on them or fitting them by maximizing the ELBo with respect to the hyperparameters in an empirical Bayes-like procedure [7], which has been shown to automatically determine appropriate levels of sparsity in other settings [58].

To fit this model, I used the mean field VI schemes described in Appendix E. Briefly, I compared the performance of the naive scheme of introducing auxiliary variables, $Y_{pk} \sim \text{Bernoulli}(1 - p_0)$, to

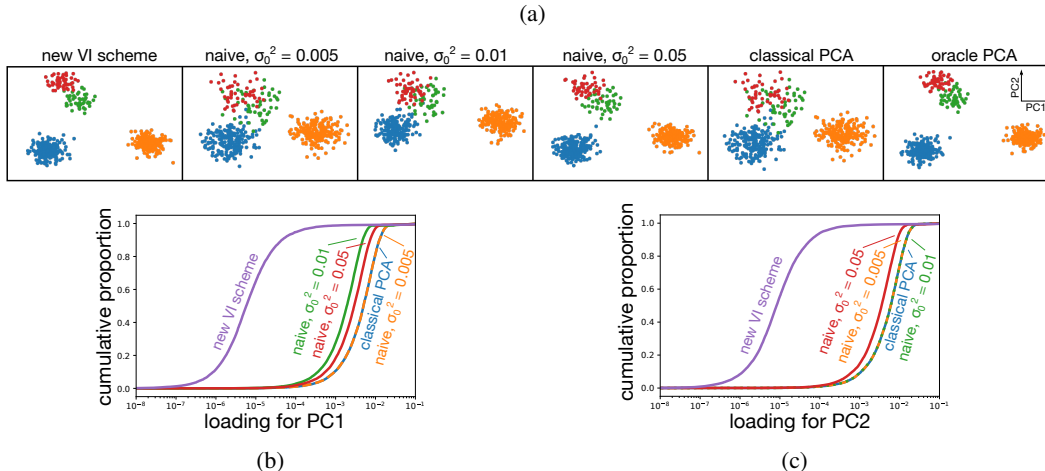

Figure 2: **Comparison of VI schemes for sparse PCA**
Data were simulated as described in the main text. Panel (a) shows the projections onto the first two PCs produced by various schemes. The results of the new VI scheme are visually comparable to that achievable by an "oracle" in this simulation scenario, while all other VI schemes do not cluster the data as well. As $\sigma_0^2$ becomes small, the naive scheme counter-intuitively recapitulates SVD. Panels (b) and (c) show that the loadings learned by the new VI scheme are sparse, while those of classical PCA and the various naive schemes are not.

split the prior on $(\mathbf{W})_{pk}$ as $(\mathbf{W})_{pk}|Y_{pk} \sim \mathcal{N}(0, \sigma^2_{Y_{pk}})$ to the scheme where the prior on $(\mathbf{W})_{pk}$ is treated exactly using the non-overlapping mixtures trick. I compared the variational posterior mean estimates of the loadings and scores from both of these schemes to classical PCA based on SVD, as well as an "oracle" version of classical PCA that uses only those variables that are simulated to have non-zero loadings.

For each dataset, I simulated 500 points with 10000 dimensions. The data was split into four clusters of sizes 200, 200, 50, and 50. For 100 of the dimensions, the value of the entry was drawn from $\mathcal{N}(\mu_c, 1)$, where $c$ indexes the cluster and each $\mu_c$ was drawn from a standard normal independently for each dimension. The remaining dimensions were drawn from standard normals. The entire matrix was then centered and scaled so that each variable had empirical mean zero, and unit empirical standard deviation, causing the simulations to differ from the generative model. For inference I set $\sigma_1^2 = 0.5$, $\sigma_e^2 = 1$, and $p_0 = 1 - 100/10000$. For all runs, I used $K = 2$ to project onto a two-dimensional space to facilitate visualization.

In the naive scheme there is an additional hyperparameter, $\sigma_0^2$, for which I consider several values. Tuning this hyperparameter is crucial to obtaining reasonable results. For $\sigma_0^2 \approx \sigma_1^2$ the model is essentially probabilistic PCA with a non-sparsity inducing Gaussian prior on the loadings, while for $\sigma_0^2 \ll \sigma_1^2$ the mean field assumption together with the zero-avoidance of VI causes the approximate posterior to put very little mass on the event $Y_{pk} = 0$, and so again the model reduces to probabilistic PCA with a Gaussian prior on the loadings. On the other hand, the VI scheme based on the non-overlapping mixtures trick produces sensible results without requiring any tuning. Indeed, Figure 2 shows that the new scheme clusters the data better than any of the naive schemes, and that as $\sigma_0^2 \downarrow 0$ the naive scheme becomes indistinguishable from classical PCA. Furthermore, whereas the posterior mean loadings from the non-overlapping mixtures trick are indeed sparse, the loadings from the other methods are dense (Figures 2b and Figures 2c). Additional simulations and a more quantitative measure of performance– reconstruction error–are presented in Appendix F. An application to a real single cell RNA-seq dataset is presented in Appendix G.

## 5   Discussion

VI has made it possible to scale Bayesian inference on increasingly complex models to increasingly massive datasets, but the error induced by obtaining an approximate posterior can be substantial [54]. Some of this error can be mitigated by using more flexible variational families, but doing so can

require alternative methods for fitting, like numerical calculation of gradients [35], sampling-based stochastic gradient estimators [34, 49], or other approximations [27]. The results of Theorems 1 and 2 provide an alternative method, using mixtures of non-overlapping exponential families to provide a more flexible variational family while maintaining conjugacy. Even in models that are not fully conjugate, methods have been developed to exploit portions of the model that are conjugate, and the results presented here may prove useful in such schemes [31]. These schemes could be especially useful in obtaining stochastic gradient updates for bayesian neural networks with spike-and-slab priors on the weights. In particular, a ReLU applied to a Gaussian random variable is a mixture of a point mass and a Gaussian restricted to be positive, which is an exponential family by Theorem 1.

Here I focused on modeling sparse phenomena and found that the non-overlapping mixtures trick is superior to a naive approach of introducing auxiliary variables. Yet, the pitfalls I described occur whenever mean field VI is applied to mixture distributions where the mixture components are very different. This suggests that in some cases, it may be beneficial to use the sparse distributions presented here to approximate non-sparse mixture distributions and then treat the sparse approximation exactly.

Throughout, I assumed that the domains of the non-overlapping mixtures were specified *a priori*. This assumption could be relaxed, treating the domains as hyperparameters that could then be optimized with respect to the ELBo. Yet, it is not obvious that for arbitrary models the objective function needs to be differentiable with respect to these hyperparameters, possibly necessitating the use of zeroth-order optimization procedures such as Bayesian Optimization [17].

Exponential families also play an important role in other forms of variational inference including Expectation Propagation [42]. The non-overlapping mixtures trick may be useful in variational approaches beyond the usual reverse KL-minimizing mean field VI.

While the non-overlapping mixtures trick makes it easy to add sparsity to conjugate models, it is not a panacea to some of the common pitfalls of VI. For example, I also considered a sparse extension of latent Dirichlet allocation (LDA) [7, 47], where documents can have exactly one topic with positive probability. Unfortunately, the zero-avoiding nature of the reverse KL results in pathological behavior: for a document with only one topic the prior topic probabilities for each word are sparse, but in the variational posterior they must be dense. Empirically, this results in the VI posterior only putting mass on the non-sparse mixture component and hence being indistinguishable from the usual VI approach to LDA. In general, care should be taken when the sparsity added to a model results in the possibility of variables having zero likelihood conditioned on latent variables coming from the sparse component.

In spite of these drawbacks, providing a recipe to easily model sparsity in otherwise conjugate Bayesian models provides another avenue to model complex phenomena while maintaining the analytical and computational benefits of mean field VI.

## Broader Impact

The primary contribution of this paper is theoretical and so the broader societal impact depends on how the theorems are used. The polygenic score application has the possibility to improve the overall quality of healthcare, but because the majority of GWAS are performed on individuals of European ancestries, PGSs are more accurate for individuals from those ancestry groups, potentially exacerbating health disparities between individuals of different ancestries as PGSs see clinical use [39]. The methods presented here are equally applicable to GWAS data collected from any ancestry group, however, and so efforts to diversify genetic data will ameliorate performance differences across ancestry groups. PGSs used for some traits such as sexual orientation [18], educational attainment [22], or stigmatized psychiatric disorders [14] raise thorny ethical considerations, especially when the application of such PGSs could enable genetic discrimination or fuel dangerous public misconceptions about the genetic basis of such traits [44]. On the other hand, PGSs applied to diseases have the potential to improve health outcomes and so if used responsibly could provide tremendous benefit to society.

## Acknowledgments and Disclosure of Funding

I would like to thank Nasa Sinnott-Armstrong and Jonathan Pritchard for piquing my interest in the polygenic score application, and Jeffrey Chan, Amy Ko, Clemens Weiss, and four anonymous reviewers for helpful feedback on the manuscript. I was funded by a National Institutes of Health training grant (5T32HG000044-23).

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
