[Supplementary Material]

# A  Analysis of naive Mean Field VI for the LDpred Model when $P = 1$

When there is only one mutation, the naive mean field VI approach to the LDpred model simplifies to

$$Z \sim \text{Bernoulli}(1 - p_0)$$
$$\beta|Z \sim \mathcal{N}(0, \sigma_Z^2)$$
$$\hat{\beta}|\beta \sim \mathcal{N}(\beta, \sigma_e^2).$$

In this simplified setting it is possible to obtain a closed form expression for the posterior. For the purposes of contrasting with VI, I consider the posterior probability that $\beta$ comes from each component of the mixture distribution, $p(Z = z|\hat{\beta})$ as well as the posterior mean of $\beta$, $\mathbb{E}[\beta|\hat{\beta}]$. In particular, for the case $\sigma_0^2 = 0$,

$$p(Z = 0|\hat{\beta}) = \frac{p_0}{p_0 + (1 - p_0)\sqrt{2\pi \frac{1}{\frac{1}{\sigma_e^2} + \frac{1}{\sigma_1^2}}} \exp\left\{\frac{\hat{\beta}^2}{2(\sigma_e^2 + \frac{\sigma_e^4}{\sigma_1^2})}\right\}} \tag{2}$$

$$\mathbb{E}[\beta|\hat{\beta}] = \left(1 - p(Z = 0|\hat{\beta})\right)\frac{\hat{\beta}}{\frac{\sigma_e^2}{\sigma_1^2} + 1}. \tag{3}$$

In this case, the usual approach to mean field VI would be to find an approximate posterior that factorizes $q(\beta, Z) = q(\beta)q(Z)$ and assume that $q(\beta)$ and $q(Z)$ are conditionally conjugate, which in this case would be that $q(\beta) = \mathcal{N}(\mu, s^2)$ and $q(Z) = \text{Bernoulli}(1 - \psi_0)$. As stated in the main text, the ELBo is undefined if $\sigma_0^2 = 0$, so instead consider $\sigma_0^2$ to be small but non-zero.

Under these assumptions, I show that for any $\hat{\beta}$ there is a $\sigma_0^2$ small enough such that the probability under $q$ that $Z = 1$ is approximately either 0 or 1 and as a result the variational posterior mean of beta is either approximately 0 or approximately equal to the non-sparse case where $p_0 = 0$. That is, mean field VI either over-shrinks effects to zero or provides no more shrinkage than just having a single gaussian prior on the effect sizes. In contrast note that $p(Z = 0|\hat{\beta})$ varies smoothly as a function of $\hat{\beta}$, and consequently by Equation 3, the posterior mean varies smoothly from shrinking tiny effects to zero to providing less shrinkage for large effects.

**Theorem 3.** *Let $q_{\hat{\beta}, \sigma_0^2}(\beta, Z)$ be the approximate posterior obtained from the LDpred model with $P = 1$ for data $\hat{\beta}$. For any $\delta$, there exists an $\epsilon$ such that for all $\sigma_0^2 < \epsilon$, either:*

$$q_{\hat{\beta}, \sigma_0^2}(Z = 0) \geq 1 - \delta \tag{4}$$

$$\left|\mathbb{E}_{q_{\hat{\beta}, \sigma_0^2}}[\beta]\right| \leq \delta \tag{5}$$

*or*

$$q_{\hat{\beta}, \sigma_0^2}(Z = 0) \leq \delta \tag{6}$$

$$\left|\mathbb{E}_{q_{\hat{\beta}, \sigma_0^2}}[\beta]\right| \geq \frac{|\hat{\beta}|}{\frac{\sigma_e^2}{\sigma_1^2} + 1} - \delta, \tag{7}$$

*with the case depending on the values of $p_0$, $\sigma_e^2$, $\sigma_1^2$, and $\hat{\beta}$.*

*Proof.* I begin by writing the ELBo:

$$\text{ELBo} = \mathbb{E}_{q_{\hat{\beta}, \sigma_0^2}}[\log p(\hat{\beta}|\beta)] - \text{KL}(q_{\hat{\beta}, \sigma_0^2}(\beta, Z)||p(\beta, Z))$$

$$= \text{constant} - \frac{1}{2\sigma_e^2}(\mu^2 + s^2 - 2\hat{\beta}\mu) + \frac{1}{2}\log s^2 - \psi_0 \log \psi_0 - (1 - \psi_0)\log(1 - \psi_0)$$

$$- \frac{1}{2}\psi_0 \log \sigma_0^2 - \frac{1}{2}(1 - \psi_0)\log \sigma_1^2 - \left(\frac{\psi_0}{2\sigma_0^2} + \frac{1 - \psi_0}{2\sigma_1^2}\right)(\mu^2 + s^2)$$

$$+ \psi_0 \log p_0 + (1 - \psi_0)\log(1 - p_0). \tag{8}$$

Taking partial derivatives I arrive at the equations for the critical points for $\mu$ and $s^2$:

$$\frac{d\text{ELBo}}{d\mu} = -\frac{\hat{\beta}}{\sigma_e^2} - \left(\frac{1}{\sigma_e^2} + \frac{\psi_0}{\sigma_0^2} + \frac{1-\psi_0}{\sigma_1^2}\right)\mu$$

$$\frac{d\text{ELBo}}{ds^2} = -\frac{1}{2\sigma_e^2} + \frac{1}{2s^2} - \frac{\psi_0}{2\sigma_0^2} - \frac{1-\psi_0}{2\sigma_1^2}.$$

Rearranging I obtain

$$\mu = \frac{\hat{\beta}}{1 + \psi_0\frac{\sigma_e^2}{\sigma_0^2} + (1-\psi_0)\frac{\sigma_e^2}{\sigma_1^2}} \tag{9}$$

$$s^2 = \frac{1}{1/\sigma_e^2 + \psi_0/\sigma_0^2 + (1-\psi_0)/\sigma_1^2}. \tag{10}$$

Now, I show that for $\psi_0 = 0$ or $\psi_0 = 1$ the ELBo is larger than for any other $\psi_0$ so long as $\lim_{\sigma_0^2 \downarrow 0} \psi_0/\sigma_0^2 > 0$. This indicates that the optimal value of $\psi_0$ must converge to either 0 or 1 in the limit of small $\sigma_0^2$ and furthermore, if $\psi_0$ converges to 0 it must do so faster than $\sigma_0^2$. Taking limits in Equations 9 and 10 under these conditions gives Equations 4, 5, 6, and 7.

If $\psi_0 = 0$, then plugging $\psi_0$ into Equations 9 and 10, it is clear that the values of both $\mu$ and $s^2$ are independent of $\sigma_0^2$. Therefore, $\text{ELBo}(\psi_0 = 0) = O(1)$.

On the other hand, plugging $\psi_0 = 1$ into Equations 9 and 10 gives $\mu = \sigma_0^2\hat{\beta}/(\sigma_0^2 + \sigma_e^2)$ and $s^2 = \sigma_0^2/(1+\frac{\sigma_0^2}{\sigma_e^2})$. Therefore, $\log s^2 = \log \sigma_0^2 - \log(1+\frac{\sigma_0^2}{\sigma_e^2}) = \log \sigma_0^2 + O(\sigma_0^2)$, and $\mu^2 + s^2 = O(\sigma_0^2)$. Plugging these results into the ELBo of Equation 8 gives

$$\text{ELBo}(\psi_0 = 1) = \frac{1}{2}\log s^2 - \frac{1}{2}\log \sigma_0^2 - \frac{1}{2\sigma_0^2}(\mu^2 + s^2) + O(1) = O(1).$$

Now, for fixed $\psi_0 \in (0,1)$,

$$\log s^2 = \log \sigma_0^2 - \log\left(\frac{\sigma_0^2}{\sigma_e^2} + \psi_0 + (1-\psi_0)\frac{\sigma_0^2}{\sigma_1^2}\right) = \log \sigma_0^2 + O(\sigma_0^2)$$

$$\mu^2 + s^2 = O(\sigma_0^2)$$

which gives an ELBo of

$$\text{ELBo}(0 < \psi_0 < 1) = \frac{1}{2}\log s^2 - \frac{1}{2}\psi_0 \log \sigma_0^2 - \frac{\psi_0}{2\sigma_0^2}(\mu^2 + \sigma^2) + O(1) = \frac{1}{2}(1-\psi_0)\log \sigma_0^2 + O(1).$$

For $\psi_0 \in (0,1)$,

$$\lim_{\sigma_0^2 \downarrow 0} \frac{1}{2}(1-\psi_0)\log \sigma_0^2 = -\infty$$

showing that in the limit of small $\sigma_0^2$, $\psi_0$ must converge to either 0 or 1. Now, because $\psi_0/\sigma_0^2$ appears in Equations 9 and 10, some care must be taken in the case where $\psi_0$ converges to 0. In particular, I show that the ELBo is larger when $\psi_0 = 0$ than it is when $\lim_{\sigma_0^2 \downarrow 0} \psi_0/\sigma_0^2 = c$, for some positive, finite constant $c$ so terms like $\psi_0/\sigma_0^2$ can be neglected in the limit when obtaining Equation 7 from Equation 9.

Noting that by Equations 9 and 10

$$\mu = \frac{s^2}{\sigma_e^2}\hat{\beta},$$

and collecting terms and rearranging the ELBo assuming that $\psi_0 < 1$ and $\psi_0 \downarrow 0$, $\sigma_0^2 \downarrow 0$, $\psi_0/\sigma_0^2 \to c$ results in

$$\text{ELBo} = -\frac{1}{2} + \frac{\hat{\beta}^2 s^2}{2\sigma_e^4} + \frac{1}{2}\log s^2 - \psi_0 \log \psi_0 - (1-\psi_0)\log(1-\psi_0)$$

$$- \frac{1}{2}\psi_0 \log \sigma_0^2 - \frac{1}{2}(1-\psi_0)\log \sigma_1^2 + \psi_0 \log p_0 + (1-\psi_0)\log(1-p_0)$$

$$= -\frac{1}{2} + \log(1-p_0) + \frac{\hat{\beta}^2 s^2}{2\sigma_e^4} + \frac{1}{2}\log s^2 + o(1).$$

Now considering the ELBo as a function of $\psi_0$, I consider the difference of the ELBo evaluated at $\psi_0 = 0$, and that evaluated at $\psi_0 < 1$ which will be denoted as $\Delta$ELBo.

$$
\begin{aligned}
\Delta\text{ELBo} &= \frac{\hat{\beta}^2}{2\sigma_e^4}(s^2|_{\psi_0=0} - s^2|_{\psi_0}) + \frac{1}{2}\log s^2|_{\psi_0=0} - \frac{1}{2}\log s^2|_{\psi_0} + o(1) \\
&> \frac{\hat{\beta}^2}{2\sigma_e^4}(s^2|_{\psi_0=0} - s^2|_{\psi_0}) + o(1) \\
&= \frac{\hat{\beta}^2}{2\sigma_e^2}\left(\frac{1}{1/\sigma_e^2 + 1/\sigma_1^2} - \frac{1}{1/\sigma_e^2 + \psi_0/\sigma_0^2 + (1-\psi_0)/\sigma_1^2}\right) + o(1) \\
&= \frac{\hat{\beta}^2}{2\sigma_e^2}\left(\frac{\psi_0/\sigma_0^2 - \psi_0/\sigma_1^2}{(1/\sigma_e^2 + 1/\sigma_1^2)(1/\sigma_e^2 + \psi_0/\sigma_0^2 + (1-\psi_0)/\sigma_1^2)}\right) + o(1) \\
&= \frac{\hat{\beta}^2}{2\sigma_e^2}\left(\frac{c}{(1/\sigma_e^2 + 1/\sigma_1^2)(1/\sigma_e^2 + c + 1/\sigma_1^2)}\right) + o(1)
\end{aligned}
$$

where the inequality follows from the fact that $s^2$ is largest when $\psi_0 = 0$. This quantity is obviously positive for any $\sigma_0^2$ sufficiently small and therefore if $\psi_0$ converges to 0, it must do so faster that $\sigma_0^2$ completing the proof. $\square$

The fact that under the VI approximate posterior $q(Z = 1)$ is either close to 0 or close to 1, while under the true posterior, $p(Z = 1|\hat{\beta})$ varies smoothly as a function of $\hat{\beta}$ suggests a thresholding phenomenon where for $\hat{\beta}$ slightly less than the threshold, the VI approximate posterior dramatically over shrinks, while for $\hat{\beta}$ slightly greater than the threshold the VI approximate posterior dramatically under shrinks essentially performing hard thresholding. In Figure 3 I show numerically that this is indeed the case, highlighting the failure of mean field VI to provide a reasonable approximation to the posterior for even this toy model.

## B    Proof of Theorem 1

*Proof.* First, note that any measure $f_{\text{mix}} \in F_{\text{mix}}$ may clearly be written as

$$
\frac{df_{\text{mix}}}{dH}(x) = \sum_{i=1}^{K} \mathbb{I}\{x \in \mathcal{S}_i\} \pi_i \exp\{\langle \eta_i, T_i\rangle - A_i(\eta_i)\} \frac{dH_i}{dH}(x),
$$

because each $f_i$ is a member of an exponential family. Since all of the events in the indicators are mutually exclusive by hypothesis, the resulting measure may be re-written as

$$
\begin{aligned}
\frac{df_{\text{mix}}}{dH}(x) &= \prod_{i=1}^{K} \exp\{\mathbb{I}\{x \in \mathcal{S}_i\}(\log \pi_i + \langle \eta_i, T_i\rangle - A_i(\eta_i))\}\left(\frac{dH_i}{dH}(x)\right)^{\mathbb{I}\{x \in \mathcal{S}_i\}} \\
&= \left[\prod_{i=1}^{K}\left(\frac{dH_i}{dH}(x)\right)^{\mathbb{I}\{x \in \mathcal{S}_i\}}\right] \exp\left\{\sum_{i=1}^{K} \mathbb{I}\{x \in \mathcal{S}_i\}(\log \pi_i + \langle \eta_i, T_i\rangle - A_i(\eta_i))\right\} \\
&= \left[\prod_{i=1}^{K}\left(\frac{dH_i}{dH}(x)\right)^{\mathbb{I}\{x \in \mathcal{S}_i\}}\right] \exp\left\{\sum_{i=1}^{K} \mathbb{I}\{x \in \mathcal{S}_i\}(\log \pi_i - A_i(\eta_i)) + \langle \eta_i, \mathbb{I}\{x \in \mathcal{S}_i\}T_i\rangle\right\} \\
&= \frac{dH_{\text{mix}}}{dH}(x)\exp\{\langle \eta_{\text{mix}}, T_{\text{mix}}(x)\rangle - A_{\text{mix}}(\eta_{\text{mix}})\}
\end{aligned}
$$

completing the proof. $\square$

## C    Proof of Theorem 2

*Proof.* Begin by noting that by the conjugacy conditions (see e.g., [16]), for any measure $f_{Y|X} \in F_{Y|X}$,

$$
df_{Y|X}(y|x) = dH^*(y)\exp\{\langle [T_Y^*(y), \alpha], T_{\text{prior}}(x)\rangle\},
$$

Figure 3: **Thresholding phenomenon for naive mean field VI for the LDpred model**
The approximate posterior mean when using naive mean field VI undergoes a thresholding phenomenon. For values of $\hat{\beta}$ close to zero, the VI scheme over-shrinks the posterior mean essentially to zero, while above the threshold, the VI scheme under-shrinks essentially matching the model without sparsity. The results here were generated with $p_0 = 0.99, \sigma_1^2 = \sigma_e^2 = 1$ and the VI model was fit using $\sigma_0^2 \approx 10^{-22}$. The results are qualitatively similar for all $\sigma_0^2 \leq 0.01$, and for larger $\sigma_0^2$ the VI model significantly under-shrinks for small $\hat{\beta}$.

for some $\alpha$ and $H^*(y)$, where $T_{\text{prior}}$ is assumed without loss of generality to be ordered in a particular way, and $T_Y^*$ are the subset of sufficient statistics of $f_{Y|X}$ that are coefficients of $T_{\text{prior}}$.

Now, since $\mathbb{I}\{x \in \mathcal{S}_i\}$ are mutually exclusive and exactly one such event occurs for each $x$,

$$df_{Y|X}(y|x) = dH^*(y)\exp\left\{\sum_{i=1}^{K}\langle[T_Y^*(y),\alpha],\mathbb{I}\{x \in \mathcal{S}_i\}T_{\text{prior}}(x)\rangle\right\}$$

$$= dH^*(y)\exp\left\{\sum_{i=1}^{K}\langle[T_Y^*(y),\alpha],\mathbb{I}\{x \in \mathcal{S}_i\}(M_iT_i(x)+v_i)\rangle\right\}$$

$$= dH^*(y)\exp\left\{\sum_{i=1}^{K}\langle M_i^T[T_Y^*(y),\alpha],\mathbb{I}\{x \in \mathcal{S}_i\}T_i(x)\rangle\right.$$

$$+ \sum_{i=1}^{K-1}(v_i^T[T_Y^*(y),\alpha]-v_K^T[T_Y^*(y),\alpha])\mathbb{I}\{x \in \mathcal{S}_i\}$$

$$\left.+ v_K^T[T_Y^*(y),\alpha]\right\}$$

where I used the hypothesis on the sufficient statistics $T_1,\ldots,T_K$ for the second equality. Multiplying by an arbitrary measure $f_{\text{mix}} \in F_{\text{mix}}$ and collecting terms I obtain

$$df_{X|Y}(x|y) \propto \exp\left\{\sum_{i=1}^{K}\langle M_i^T[T_Y^*(y),\alpha]+\eta_i,\mathbb{I}\{x \in \mathcal{S}_i\}T_i(x)\rangle\right.$$

$$+ \sum_{i=1}^{K-1}\left(v_i^T[T_Y^*(y),\alpha]-v_K^T[T_Y^*(y),\alpha]\right.$$

$$\left.\left.+ \log\pi_i - A_i(\eta_i) - \log\pi_K + A_K(\eta_K)\right)\mathbb{I}\{x \in \mathcal{S}_i\}\right\}dH_{\text{mix}}(x)$$

$$= \exp\left\{\langle\eta^*_{\text{mix}},T_{\text{mix}}(x)\rangle\right\}dH_{\text{mix}}(x),$$

where $\eta^*_{\text{mix}}$ is the updated parameter obtained by collecting terms, showing that the posterior is in the same exponential family as the prior. $\square$

## D    VI Schemes for the LDpred model

Recall that the naive VI scheme introduces auxiliary variables to approximately model sparsity as in Equation 1. The natural mean field approach would then be to approximate the posterior over $\beta_j$ as a Gaussian with mean $\mu_j$ and variance $s_j^2$, and approximate the posterior over $Z_j$ as a Bernoulli with probability of being zero $\psi_j$ to maintain conditional conjugacy.

Routine calculations then show that the coordinate ascent updates are

$$\psi_i \leftarrow \frac{p_0\exp\left\{-\frac{1}{2}\log\sigma_0^2 - \frac{1}{2\sigma_0^2}(\mu_i^2+s_i^2)\right\}}{p_0\exp\left\{-\frac{1}{2}\log\sigma_0^2 - \frac{1}{2\sigma_0^2}(\mu_i^2+s_i^2)\right\}+(1-p_0)\exp\left\{-\frac{1}{2}\log\sigma_1^2 - \frac{1}{2\sigma_1^2}(\mu_i^2+s_i^2)\right\}}$$

$$\mu_i \leftarrow \frac{\hat{\beta}_i - \sum_{j\neq i}\mathbf{X}_{ij}\mu_j}{\psi_i\sigma_e^2/\sigma_0^2 + (1-\psi_i)\sigma_e^2/\sigma_1^2 + \mathbf{X}_{ii}}$$

$$s_i^2 \leftarrow \frac{1}{\psi_i/\sigma_0^2 + (1-\psi_i)/\sigma_1^2 + \mathbf{X}_{ii}/\sigma_e^2}.$$

Using Theorems 1 and 2 it is possible to derive an alternative VI scheme that eschews the need for auxiliary variables. In particular, the set of distributions containing only the point mass at 0 is trivially an exponential family, and the support of distributions in that family do not overlap with the set of Gaussians supported on $\mathbb{R}\setminus\{0\}$. Therefore, the set of distributions that are mixtures of a Gaussian and a point mass at 0 are also an exponential family by Theorem 1. Then, by Theorem 2, because a Gaussian prior on the mean of a Gaussian is conjugate, and the sufficient statistics of a Gaussian are

constant on the set $\{0\}$, this mixture distribution is also a conjugate prior for the mean of a Gaussian. The natural approximation to make for the variational posterior over $\beta_j$ would then lie in the same exponential family – a mixture of a Gaussian with mean $\mu_j$ and variance $s_j^2$ and a point mass at 0, with the probability of 0 being $\psi_j$.

Because the model is conjugate and the distributions are in the exponential family, the optimal updates for the natural parameters can be obtained from

$$q(\beta_i) \propto \exp\left\{ \mathbb{E}_{-i} \log P(\hat{\beta}|\beta) + \log P(\beta_i) \right\} \tag{11}$$

where $\mathbb{E}_{-i}[\cdot]$ is short hand for taking the expectation under the approximate posterior with respect to all variables except $\beta_i$. The posterior mean under the variational approximation is $\mathbb{E}_q[\beta_i] = (1-\psi_i)\mu_i$, and so the first term expands to

$$\mathbb{E}_{-i} \log P(\hat{\beta}|\beta) = -\frac{1}{2\sigma_e^2} \mathbb{E}_{-i}(\hat{\beta} - \mathbf{X}\beta)^T \mathbf{X}^{-1}(\hat{\beta} - \mathbf{X}\beta)$$

$$= -\frac{1}{2\sigma_e^2} \mathbf{X}_{ii}\beta_i^2 + \frac{1}{\sigma_e^2}\left( \hat{\beta}_i - \sum_{j \neq i} \mathbf{X}_{ij}(1-\psi_j)\mu_j \right)\beta_i + \text{const}$$

$$= -\frac{1}{2\sigma_e^2} \mathbf{X}_{ii}\mathbb{I}\{\beta_i \neq 0\}\beta_i^2 + \frac{1}{\sigma_e^2}\left( \hat{\beta}_i - \sum_{j \neq i} \mathbf{X}_{ij}(1-\psi_j), \mu_j \right)\mathbb{I}\{\beta_i \neq 0\}\beta_i + \text{const}.$$

The natural parameters for a Gaussian with mean $\mu_i$ and variance $s_i^2$ are $-\frac{1}{2s_i^2}$ and $\frac{\mu_i}{s_i^2}$ with log normalizer $\frac{\mu_i^2}{2s_i^2} + \frac{1}{2}\log s_i^2$, with corresponding sufficient statistics $\beta_i^2$ and $\beta_i$. By Theorem 1, the natural parameters for the mixture distribution are therefore $-\frac{1}{2s_i^2}$, $\frac{\mu_i}{s_i^2}$, and $\log \psi_i - \log(1-\psi_i) + \frac{\mu_i^2}{2s_i^2} + \frac{1}{2}\log s_i^2$, with corresponding sufficient statistics $\mathbb{I}\{\beta_i \neq 0\}\beta_i^2$, $\mathbb{I}\{\beta_i \neq 0\}\beta_i$, and $\mathbb{I}\{\beta_i = 0\}$. Matching the coefficients of the sufficient statistics in Equation 11 and performing some algebra produces

$$\psi_i \leftarrow 1 - \frac{1}{1 + \frac{p_0}{1-p_0}\sqrt{1 + \mathbf{X}_{ii}\sigma_1^2/\sigma_e^2} \exp\left\{ \frac{-\left( \hat{\beta}_i - \sum_{j \neq i} \mathbf{X}_{ij}\mu_j(1-\psi_j) \right)^2}{2\sigma_e^4/\sigma_1^2 + 2\sigma_e^2\mathbf{X}_{ii}} \right\}}$$

$$\mu_i \leftarrow \frac{\hat{\beta}_i - \sum_{j \neq i} \mathbf{X}_{ij}\mu_j(1-\psi_j)}{\sigma_e^2/\sigma_1^2 + \mathbf{X}_{ii}}$$

$$s_i^2 \leftarrow \frac{1}{1/\sigma_1^2 + \mathbf{X}_{ii}/\sigma_e^2}.$$

When fitting either VI scheme, I performed 100 iterations of coordinate ascent using the above update. For the naive scheme, for coordinate $i$, I update $\mu_i$ and $s_i^2$ first, then update $\psi_i$ before moving on to coordinate $i+1$. For initialization, $\mu_i = 0$ for all $i$, and $s_i^2 = \sigma_1^2 + \sigma_e^2$. For the naive case, $\psi_i$ was initialized to be 1, while for new scheme, $\psi_i$ was initialized to be $p_0$.

In both VI schemes, the rate-limiting step is clearly computing terms that involve summations of the type $\sum_{j \neq i}$, which take $O(P)$ time, where $P$ is the number of variables. Since there are $O(P)$ variational parameters to update at each iteration, the runtime of each iteration is thus $O(P^2)$.

# E VI Schemes for sparse probabilistic PCA

First I derive the naive VI scheme. For the auxiliary model,

$$Y_{pk} \sim \text{Bernoulli}(1 - p_0)$$
$$\mathbf{W}_{pk}|Y_{pk} \sim \mathcal{N}(0, \sigma^2_{Y_{pk}})$$
$$Z_n \sim \mathcal{N}(0, \mathbf{I}_K)$$
$$X_n|Z_n, \mathbf{W} \sim \mathcal{N}(\mathbf{W}Z_n, \sigma^2_e \mathbf{I}_P)$$

The natural mean field VI scheme for this model would be to assume that all variables are independent and assume that under the posterior $Y_{pk}$ is Bernoulli with parameter $\psi_{pk}$, $\mathbf{W}_{pk}$ is Gaussian with mean $\mu_{W_{pk}}$ and variance $s^2_{W_{pk}}$, and $Z_n$ is multivariate normal with mean $\mu_{Z_n}$ and covariance matrix $\mathbf{S}_{Z_n}$. Below, I use the notation

$$\mathbf{X} := \begin{pmatrix} | & & | \\ X_1 & \cdots & X_N \\ | & & | \end{pmatrix}.$$

Routine calculations result in the following updates:

$$\mu_{Z_n} \leftarrow \frac{1}{\sigma^2_e} \left( \frac{1}{\sigma^2_e} \mathbb{E}[\mathbf{W}^T\mathbf{W}] + \mathbf{I}_K \right)^{-1} \mathbb{E}[\mathbf{W}]^T X_n$$

$$\mathbf{S}_{Z_n} \leftarrow \left( \frac{1}{\sigma^2_e} \mathbb{E}[\mathbf{W}^T\mathbf{W}] + \mathbf{I}_K \right)^{-1}$$

$$s^2_{W_{pk}} \leftarrow \left[ \frac{1}{\sigma^2_e} \left( \sum_n \mu^2_{Z_n,k} + \mathbf{S}^2_{Z_n,kk} \right) + \frac{\psi_{pk}}{\sigma^2_0} + \frac{1 - \psi_{pk}}{\sigma^2_1} \right]^{-1}$$

$$\mu_{W_{pk}} \leftarrow \frac{s^2_{W_{pk}}}{\sigma^2_e} \left[ \left( \sum_n \mathbf{X}_{np} \mu_{Z_n,k} \right) - \left( \sum_n \sum_{\ell \neq k} \mu_{W_{p\ell}} \mathbf{S}_{Z_n,k\ell} \right) \right]$$

$$\psi_{pk} \leftarrow 1 - \frac{1}{1 + \frac{p_0}{1-p_0} \sqrt{\sigma^2_1/\sigma^2_0} \exp\left\{ \frac{1}{2} \left( \frac{1}{\sigma^2_0} - \frac{1}{\sigma^2_1} \right) \left( \mu^2_{W_{pk}} + s^2_{W_{pk}} \right) \right\}}$$

where

$$\mathbb{E}[\mathbf{W}]_{pk} = \mu_{W_{pk}}$$

and

$$\mathbb{E}[\mathbf{W}^T\mathbf{W}]_{k\ell} = \sum_p \mu_{W_{pk}} \mu_{W_{p\ell}} + \delta_{k\ell} \sum_p s^2_{W_{pk}}.$$

Now I derive a VI scheme using Theorems 1 and 2. The calculations are largely the same as in Appendix D and so a number of details are omitted. Because I am again replacing a Gaussian by a mixture of a Gaussian and point mass at zero, I assume the posterior for $\mathbf{W}_{pk}$ is a mixture of a point mass at zero and a Gaussian with mean $\mu_{W_pk}$, variance $s^2_{W_{pk}}$, and probability of being zero

$\psi_{pk}$. Working through the algebra as in the LDpred model results in:

$$\mu_{Z_n} \leftarrow \frac{1}{\sigma_e^2} \left( \frac{1}{\sigma_e^2} \mathbb{E}[\mathbf{W}^T\mathbf{W}] + \mathbf{I}_K \right)^{-1} \mathbb{E}[\mathbf{W}]^T X_n$$

$$\mathbf{S}_{Z_n} \leftarrow \left( \frac{1}{\sigma_e^2} \mathbb{E}[\mathbf{W}^T\mathbf{W}] + \mathbf{I}_K \right)^{-1}$$

$$s_{W_{pk}}^2 \leftarrow \left[ \frac{1}{\sigma_e^2} \left( \sum_n \mu_{Z_n,k}^2 + \mathbf{S}_{Z_n,kk}^2 \right) + \frac{1}{\sigma_1^2} \right]^{-1}$$

$$\mu_{W_{pk}} \leftarrow \frac{s_{W_{pk}}^2}{\sigma_e^2} \left[ \left( \sum_n \mathbf{X}_{np}\mu_{Z_n,k} \right) - \left( \sum_n \sum_{\ell \neq k} \mu_{W_{p\ell}}(1 - \psi_{p\ell})\mathbf{S}_{Z_n,k\ell} \right) \right]$$

$$\psi_{pk} \leftarrow 1 - \frac{1}{1 + \frac{p_0}{1-p_0}\sqrt{\sigma_1^2/s_{W_{pk}}^2}\exp\left\{ -\frac{\mu_{W_{pk}}^2}{2s_{W_{pk}}^2} \right\}}$$

where

$$\mathbb{E}[\mathbf{W}]_{pk} = \mu_{W_{pk}}(1 - \psi_{pk})$$

and

$$\mathbb{E}[\mathbf{W}^T\mathbf{W}]_{k\ell} = \sum_p \mu_{W_{pk}}(1 - \psi_{pk})\mu_{W_{p\ell}}(1 - \psi_{p\ell}) + \delta_{k\ell}\sum_p s_{W_{pk}}^2(1 - \psi_{pk}).$$

When fitting both VI schemes, I performed 250 iterations of coordinate ascent. For the naive scheme, I first updated every coordinate of $Z$, then for each coordinate updated $Y_{pk}$ then $\mathbf{W}_{pk}$. For the new scheme, I first updated $Z$ coordinate-wise then updated $\mathbf{W}$ coordinate-wise. Using singular value decomposition to decompose $\mathbf{X} = \mathbf{U}\boldsymbol{\Sigma}\mathbf{V}^T$, I initialized $\mu_{Z_i} = \mathbf{U}_n$, $\mathbf{S}_{Z_n} = \mathbf{I}_2$ $\mu_{W_{pk}} = \mathbf{V}_{pk}\boldsymbol{\Sigma}_{kk}$, $s_{W_{pk}}^2 = 1$ and $\psi_{pk} = 1 \times 10^{-10}$ for both schemes.

The updates for both models require the inversion of a $K \times K$ matrix which is $O(K^3)$ and computing $\mathbb{E}[\mathbf{W}^T\mathbf{W}]$ is $O(PK^2)$, but these can be precomputed before updating each $\mu_{Z_n}$ and $\mathbf{S}_{Z_n}$. Then, updated each $\mu_{Z_n}$ requires $O(K^2 + PK)$ time. Therefore, updating all $\mu_{Z_n}$ and $\mathbf{S}_{Z_n}$ requires $O(NPK)$ time assuming that $K \ll N$ and $K \ll P$. For fixed $\mu_{Z_n}$ and $\mathbf{S}_{Z_n}$, updating $\mu_{W_{pk}}$, $s_{W_{pk}}^2$, and $\psi_{pk}$ is limited by computing $\sum_n \sum_{\ell \neq k} \mu_{W_{p\ell}}\mathbf{S}_{Z_n,k\ell}$ or $\sum_n \sum_{\ell \neq k} \mu_{W_{p\ell}}(1 - \psi_{W_{p\ell}})\mathbf{S}_{Z_n,k\ell}$ which requires $O(NK)$ time. Therefore updating all $\mu_{W_{pk}}$, $s_{W_{pk}}^2$, and $\psi_{pk}$ requires $O(NPK^2)$ time. Therefore, each iteration of coordinate ascent requires $O(NPK^2)$ time.

# F  Additional PCA runs

To ensure that the results presented in the main text are not unusual, I generated five additional datasets as described in the main text and compared the resulting PCA projections and sparsity of the loadings for traditional PCA (based on singular value decomposition), my naive implementation of sparse probabilistic PCA, and the implementation of sparse probabilistic PCA based on the non-overlapping mixtures trick (Figures 4 and 5). In all five realizations, the new formulation of sparse probabilistic PCA produces the sparsest loadings, and subjectively best separates the four clusters using the first two principle components. As before, the naive implementation is indistinguishable from traditional PCA for small values of $\sigma_0^2$ or values of $\sigma_0^2$ close to 1.

I also computed reconstruction error as a quantitative measure of performance. I defined reconstruction error as the squared Frobenius norm between the reconstructed matrix ($\mathbb{E}[\mathbf{WZ}]$ for the VI methods) and the signal in the simulated matrix – that is, the matrix obtained by centering and scaling a matrix where each entry is the $\mu_c$ for that entry as defined above. The mean reconstruction error across five simulations was 4261 (min: 4072, max: 4517) for the non-overlapping mixtures trick; 3985 (min: 3923, max: 4240) for oracle PCA; and 29407 (min: 28761, max: 30429) for classical PCA. Across the naive schemes, taking $\sigma_0^2 = 0.05$ performed best with a mean reconstruction error of 7191 (min:7090, max: 7408). Overall, the non-overlapping mixtures trick performed only slightly worse than PCA using knowledge of which variables were non-zero, whereas even the best naive

Figure 4: **Projections onto the first two PCs for five replicate simulations**
Across all five replicate simulations the data cluster well in their projection onto the first two PCs for the new VI scheme. While the naive scheme can cluster the data well if $\sigma_0^2$ is tuned properly, the clusters are often not as well-defined as under the new scheme. Furthermore, the loadings are substantially less sparse as shown in Figure 5. In the limit of $\sigma_0^2 \downarrow 0$, it empirically appears that the naive scheme is indistinguishable from classical PCA.

scheme had almost double the reconstruction error, but all methods that attemted to account for sparsity outperformed classical PCA.

# G Application of sparse probabilistic PCA to single cell RNA-seq data

To provide an application of the non-overlapping mixtures trick to a real dataset, I applied the VI scheme derived above for the sparse probabilistic PCA model with a spike-and-slab prior on the loadings to a single cell RNA-seq dataset.

In a simplified view of the cell, genes encoded in a cell's DNA are transcribed into RNA, which in turn is translated into proteins. Proteins perform the functions necessary for maintaining the cell as well as performing the specialized tasks that different cell types perform in complex multicellular organisms, such as humans. The DNA sequence is–for the most part–identical across the cells of a single organism, yet different cells perform highly differentiated and specialized roles. To understand this process of differentiation, scientists isolate single cells and sequence their RNA, providing a snapshot of the relative abundances of the RNA coding different genes. The result of these experiments are summarized by a cell by gene matrix containing the abundance of the RNA from each gene in each cell.

I analyzed a single cell RNA-seq dataset[1] of peripheral blood mononuclear cells (PBMCs) – a subset of the "white blood cells" circulating in blood. The RNA-seq data were processed and cleaned using `kallisto/bustools` [40] and `scanpy` [59] as described in the `kallisto` tutorial[2]. I performed standard PCA as well as sparse probabilistic PCA using the non-overlapping mixtures trick, obtaining "gene loadings" and PC scores for the individual cells. The mean posterior probabilities of the gene loadings being non-zero are plotted in Figure 6 – for all loadings, sparse PCA only has appreciable posterior probabilities of being non-zero for less than a third of the genes. Yet, as seen in Figure 7 the cells cluster similarly as classical PCA. Additionally, known markers of different PBMC cell-type lineages, such as the GATA3, SPI1, and BCL2 [3, 13, 9] are highly differentiated across the

Figure 5: **Distribution of loadings for five replicate simulations**
The new VI scheme produces significantly sparser loadings, with about 90% of variables having an absolute loading below $1 \times 10^{-5}$ for both PC1 and PC2. The naive scheme, while having a more skewed distribution of loadings than classical PCA can hardly be considered sparse with most variables having loadings greater than $1 \times 10^{-3}$ regardless of the precise value of $\sigma_0^2$ used. Note that in both plots, the right-most cluster of curves is over-plotted: for PC1 the naive scheme with $\sigma_0^2 = 0.005$ is essentially indistinguishable from classical PCA for all five replicates. For PC2, the naive scheme with $\sigma_0^2 = 0.005$ or $\sigma_0^2 = 0.01$ is indistinguishable from classical PCA in all but one replicate.

Figure 6: **Average posterior probability of loadings being non-zero**
For the single cell PBMC dataset, posterior inclusion probabilities (PIPs) – that is the probability of a loading being non-zero – were computed for each loading for each PC, and the average across genes for each PC is presented here. All components have average PIPs lower than 0.3 suggesting that less than one third of genes are required to obtain a meaningful low-dimensional representation of the cells.

clusters, all of which have essentially probability one of having non-zero loadings on the first PC under the posterior. Finally, the posterior means under the sparse model are not merely a monotone transformation of the classical PCA loadings, as seen in Figure 8, suggesting that sparse PCA may prioritize different genes for followup than classical PCA.

Figure 7: **Projections and expression of marker genes**
The left column shows each cell's PC scores of the first two PCs from performing PCA on the PBMC data using classical PCA. The right column shows the PC scores from running the VI scheme derived using the non-overlapping mixtures trick for sparse probabilistic PCA. In each row, the expression level of a different marker gene is shown – purple is lowly expressed, and yellow is highly expressed. In both PCA and sparse PCA, different cell types are well-clustered, with marker gene expression diverging across clusters.

Figure 8: **Posterior mean loadings compared to classical PCA loadings**
The posterior mean from the sparse probabilistic PCA model of each loading for PC 1 (left) and PC 2 (right) are compared to the loadings from classical PCA for the PBMC dataset. A large number of genes that have non-zero loadings under classical PCA are shrunk to nearly zero in the sparse PCA posterior mean. Furthermore, the sparse PCA loadings are not merely a monotonic transformation of the classical PCA loadings – the relative ordering of genes changes somewhat. This suggests that different genes may be prioritized for followup using the two methods.

## Footnotes

[1] https://support.10xgenomics.com/single-cell-gene-expression/datasets/3.0.0/pbmc_1k_v3

[2] https://colab.research.google.com/github/pachterlab/kallistobustools/blob/master/notebooks/kb_analysis_0_python.ipynb