[Reviews · NeurIPS 2020]

Review 1

Summary and Contributions: This paper extends the usefulness of variational inference by showing that more flexible families of exponential families can be constructed piecewise as long as the support of the sub-distributions do not overlap. This idea is developed into a way to build sparse models with spike-and-slap priors to which mean-field variational inference can be applied. The paper considers in detail two use cases: gene-wide association studies (GWAS) and sparse probabilistic PCA. Simulation experiments, partially based on real data, show the method works effectively. Update: I have read the author rebuttal and am satisfied that it will be possible to add the additional comparisons and glad the author is able to add further empirical comparisons to BBBVI and MCMC. The new title is also an improvement!

Strengths: The theoretical claim that you can always form an exponential family out of mixtures of other exponential families in a piecewise way is very intriguing and a flexible tool that could be used in other settings, e.g., expectation propagation as the author mentions. The motivation was compelling: imbuing our models with sparsity often makes inference harder, so here is a way to pose the inference problem as a variational inference procedure on sparse models using conjugate exponentials. The motivating case of gene-wide association studies (GWAS) was strong and helped bring concreteness to the theoretical idea. It was also very interesting to learn more about GWAS in the process, there was enough detail to bring the problem to life. I also appreciated the paragraph in Section 5 on the failure case of a sparse version of LDA where documents can have exactly one topic with positive probability. In this case, the method seems to revert to standard LDA but it is good to have such guidelines to avoid self-contradiction about the possibility of sparsity in the posterior.

Weaknesses: The gap in the paper that I did not see addressed was structured variational inference. If the problem of combining sparse models with mean-field variational inference is that the sparsity membership gets decoupled from the other parameters, then avoiding making them independent in the variational family would address this. Note, this is distinct from the black-box variational inference or VAE approaches as in some cases of structured VI you can still have closed-form updates that do not rely on the REINFORCE gradient or the reparameterization trick.

Correctness: Theoretical and empirical contributions are correct.

Clarity: Paper is very clearly written throughout. Here are some specific suggestions for improvement. Reference to MCMC as a solution to sparse models inference is made in the paper, particularly the paragraph starting on line 104, but there is not much detail on the points of comparison with VI. Is it really the case that MCMC performs well on spike and slab models and the only drawback is computational speed? specific notes: line 94: the "... however ..." seems out of place in text even though it would make sense in speech line 160: "form" -> "from"

Relation to Prior Work: Related work is clearly discussed. Citations to related GWAS studies were particularly helpful. Here are some references to structured variational inference: http://papers.nips.cc/paper/1155-exploiting-tractable-substructures-in-intractable-networks.pdf http://proceedings.mlr.press/v38/hoffman15.html

Reproducibility: Yes

Additional Feedback: I have a question: is the method always constrained to define the support before running inference or is it possible to make the support flexible, perhaps as a hyperparameter that could be fit in a separate procedure? My humble suggestion is to change the title of the paper to something more descriptive. It is true that the paper is about "sparsity, conjugacy, and mean field variational inference" but a title that is more descriptive than a bag of words might generate more interest.


Review 2

Summary and Contributions: [Update] I have carefully read the authors' response and the other reviews. My primary concern with the original submission was the empirical validation. The additional experiments reported in the authors' response represent, I think, a significant improvement and I believe that their incorporation into a revised version of the paper will substantially strengthen the final paper. For this reason I am raising my score to a 7 and would be happy to see this work accepted for publication. ======================= The author shows that mixtures of exponential family distributions with non-overlapping support form an exponential family. This observation is then used to define mean field variational inference algorithms with closed form updates for a particular class of conditionally conjugate models. The focus is on models with sparsity-inducing priors such as the spike-and-slab. Empirical results demonstrate that the resulting inference algorithm can outperform vanilla mean field VI.

Strengths: - theorems 1 & 2 and their application to VI are, to the best of my knowledge, novel. it seems likely that this "trick" could be more broadly applicable. - experiments provide some evidence that the proposed VI scheme outperforms vanilla VI alternatives.

Weaknesses: - the experiments are somewhat cursory, considering only two particular models and limited entirely to simulated data. - the exposition could be improved. in particular more care could be taken to define notation and large contiguous blocks of text could be reorganized for readability.

Correctness: I have no concerns about the correctness of the methodology.

Clarity: The paper is reasonably well-written, although I would argue that the exposition could be improved in a number of respects. - more care could be taken to define notation (e.g. the dimensionality of X in lines 98/99) - there are several places where additional comments could guide reader intuition. for example it would be useful to: i) unpack Thm 2 in more detail; and ii) comment on the asymmetry in Thm 1 wrt component K.

Relation to Prior Work: While there is some discussion of prior work it might be useful to broaden the discussion to include relations to other work in the literature, especially more recent work that is more "black-box" in nature. Suggestions that come to mind (there may be better ones) include: [1] "Approximating exponential family models (not single distributions) with a two-network architecture," Sean R. Bittner, John P. Cunningham. [2] "Composing graphical models with neural networks for structured representations and fast inference." Matthew J. Johnson, David K. Duvenaud, Alex Wiltschko, Ryan P. Adams, Sandeep R. Datta While these sorts of work are different to the present work in many respects, they are similar in the sense that they both explore the theme of "how far can we get by exploiting exponential family/conjugate structure?"

Reproducibility: Yes

Additional Feedback: Broadly, I think this work could benefit from additional empirical evidence to better make the case for the practicality and usefulness of the proposed methods. This might include considering an additional class of models amenable to the "non-overlapping mixtures trick"; it would also certainly include experiments that go beyond simulated data. Miscellaneous: - It might be useful to include reconstruction error in the metrics considered in the sparse PCA experiment. - It would be useful to comment on the form of the variance in the likelihood in the LDpred model. Typos: line 125: p->P line 160: and -> an line 238: posterior -> posteriori


Review 3

Summary and Contributions: The paper shows an interesting result that a spike-and-slab prior with the slab belonging to an exponential family also belongs to an exponential family. Mean field VB is less accurate if it requires a deep Bayesian hierarchy using too many auxiliary variables to help its derivation. This conjugacy discovery might help to ease derivation of mean-field variational Bayes algorithms for variable selection in regression problems with a spike-and-slab prior.

Strengths: Please see the summary.

Weaknesses: Theorem 1 is nice and its proof is correct. However, it is not clear how the result still holds for a spike-and-slab distribution as the supports of spike and slap overlap. As the application to the spike-and-slab distribution is the main focus of the paper, I would ask the author to make this clear, e.g., by adding a corollary to show the conjugacy of the spike-and-slab distribution. Also, it would be nice if the author rephrases Theorem 2 a bit to make it easier to understand; its current statement is not clear how the result can be used. A toy example explaining its use would be nice.

Correctness: Yes

Clarity: Yes

Relation to Prior Work: Yes

Reproducibility: Yes

Additional Feedback:


Review 4

Summary and Contributions: Sparsity is important in most applications as it increases the interpretability of the models. Sparsity can be applied to the models by assuming applying sparse priors to the model parameters such as spike-and-slab. In such scenarios for obtaining the posteriors there are two approaches: MCMC methods and variational inference (VI). Mean field VI is preferred in most cases because of simplicity and computational costs. However, this paper shows that using standard mean field VI when there is sparsity in priors yields suboptimal performance. As a results, this paper introduces the non-overlapping mixtures trick by providing two theorems. This means that mixtures of exponential family distributions with nonoverlapping support form an exponential family. In particular, any mixture of a diffuse exponential family and a point mass at zero to model sparsity forms an exponential family. So, this method can simultaneously improve the flexibility of the inference and keep the computational cost low.

Strengths: In variational inference (VI), there is always a tradeoff between flexibility of variational posterior and computational costs. This paper proposes a new VI scheme, when there is sparsity, bu the idea of non-overlapping mixtures trick. The strength of the paper is providing two useful theorems which proves this idea that mixtures of exponential family distributions with nonoverlapping support form an exponential family. This actually lays the ground for efficient inference of posteriors when we use sparse priors like spike-and-slab.

Weaknesses: One weakness of the paper is that there is no real-world experiments. Another weakness is lack of comparisons with other more advanced VI methods.

Correctness: They seem to be correct.

Clarity: Yes

Relation to Prior Work: Yes.

Reproducibility: Yes

Additional Feedback: The idea of non-overlapping mixtures trick is interesting. Theoretical grounds of the idea in context of two theorems are also promising. However, some issues that comes to my mind are: 1- there is not a real-world experiments in the paper. Adding some real experiments could enhance the value of the paper, 2- in all the synthetic experiments, the comparisons were done only with standard VI methods, but there are recent advanced VI methods which also consider more expressive families of posteriors such as SIVI (semi-implicit variational inference) or normalizing flow. I was wondering how the proposed method could compare with those methods on a given task? The response has been read.

[Author Response · NeurIPS 2020]

I would like to thank the reviewers for their thorough and constructive comments, and I am confident that all of the
feedback can be easily incorporated in a revision. The comments broadly fell into three areas, which I address below: 1)
an application of the non-overlapping mixtures trick (NOMT) to real data; 2) a comparison of the NOMT to a broader
class of methods for obtaining approximate posteriors; and 3) changes to the exposition.

**Real Data Application:** While the primary result of this work is theoretical, I agree with Reviewers 2 and 4 that
including an application to real data would strengthen the manuscript. To that end, I performed an analysis of a
publicly available single cell RNA-seq dataset of peripheral mononuclear blood cells (PMBCs) from 10X Genomics
("pbmc_1k_v3") using the NOMT for sparse PCA. Sparse PCA better separates known cell types with sparser loadings
than classical PCA (using only about 12% as many genes). Furthermore, by having an explicitly sparse model, the
posterior probability that a feature is non-zero can be interpreted as a Posterior Inclusion Probability (PIP). Genes with
high PIP in the first few PCs include known markers of immune cell lineage.

**Comparison to other methods:** To further highlight the utility of the NOMT, I compared the VI scheme for the
LDpred model (coming from GWAS) based on the NOMT to boosting black-box VI (BBBVI) [3] as implemented in
pyro [1] and to adaptive random walk MCMC as implemented in NIMBLE [2]. I simulated 20 datasets as described in
the manuscript at each level of noise: $\sigma_e^2 \in \{1.0, 0.5, 0.1, 0.05\}$. For BBBVI, I used independent Gaussians for each $\beta_j$
and independent Bernoullis for each $Z_j$ as the variational family, and did 5 iterations of boosting – allowing mixtures of
5 distributions from the variational family. NIMBLE was run for 1000 passes over the data which took about 12 hours
per dataset. BBBVI took about 3 hours per dataset. The NOMT took less than 5 seconds per dataset. The results are
presented below (mean across simulations $\pm$ two standard errors): – the NOMT performs almost identically to MCMC
by the metrics considered, while BBBVI performs substantially worse.

| Method | MSE $\times 1000$ | | | | Correlation | | | |
|---|---|---|---|---|---|---|---|---|
| | $\sigma_e^2 = 1.0$ | $\sigma_e^2 = 0.5$ | $\sigma_e^2 = 0.1$ | $\sigma_e^2 = 0.05$ | $\sigma_e^2 = 1.0$ | $\sigma_e^2 = 0.5$ | $\sigma_e^2 = 0.1$ | $\sigma_e^2 = 0.05$ |
| MCMC | $9.1 \pm 0.4$ | $8.2 \pm 0.3$ | $2.7 \pm 0.2$ | $1.5 \pm 0.1$ | $0.16 \pm 0.02$ | $0.33 \pm 0.03$ | $0.84 \pm 0.01$ | $0.89 \pm 0.01$ |
| BBBVI | $9.8 \pm 0.5$ | $9.0 \pm 0.4$ | $9.1 \pm 0.5$ | $8.7 \pm 0.5$ | $0.08 \pm 0.00$ | $0.11 \pm 0.00$ | $0.23 \pm 0.01$ | $0.29 \pm 0.01$ |
| NOMT | $9.1 \pm 0.4$ | $8.2 \pm 0.3$ | $2.7 \pm 0.2$ | $1.5 \pm 0.1$ | $0.16 \pm 0.02$ | $0.34 \pm 0.03$ | $0.84 \pm 0.01$ | $0.89 \pm 0.01$ |

**Changes to the exposition:**

Reviewer 3 noted that the spike-and-slab model does not satisfy the non-overlapping support assumption of Theorem 1.
For any non-atomic distribution such as the Gaussian component in the spike-and-slab we may remove a set of measure
zero without changing the density. We may therefore define an exponential family with the same density but supported
on $\mathbb{R} \setminus \{0\}$, which does not overlap the point mass at zero.

Reviewer 2 pointed out that there is an interesting asymmetry in Theorem 1 with respect to component $K$. This arises
because the mixture weights are constrained to sum to one, so the $K^{\text{th}}$ mixture weight is completely determined by
the other mixture weights. It would be possible to have a "symmetric" version of the theorem, but it would describe a
curved exponential family and many downstream results pertain only to exponential families that are not curved.

Reviewer 2 suggested using reconstruction error as a metric for the sparse PCA application. I have computed
reconstructions errors for 5 more replicate simulations as described in the manuscript. I computed the reconstruction
error for the "signal" in the matrix, that is the matrix before gaussian noise added to each element. The mean
reconstruction error for the NOMT was 4,261; for the best naive VI scheme it was 7,191; for standard PCA it was
29,407; and for the "oracle" version of PCA it was 3,985.

Reviewers 1 and 3 pointed out additional methods for fitting approximate posteriors: structured VI and a number of
black-box VI methods. While many of these methods are similar in spirit, they apply to different types of models than
the present work. I will include a discussion of these similarities and differences in the revision.

Reviewer 1 asked if the supports of the mixture distributions must defined *a priori*. They could certainly be treated as
hyper-parameters and optimized to maximize the ELBo in a variational analog of Empirical Bayes.

Reviewer 1 suggested changing the title as it is currently too broad. I agree, and will change the title to "Flexible mean
field variational inference using mixtures of non-overlapping exponential families."

[1] Eli Bingham et al. Pyro: Deep universal probabilistic programming. *The Journal of Machine Learning Research*,
20(1):973–978, 2019.

[2] Perry de Valpine et al. Programming with models: writing statistical algorithms for general model structures with
NIMBLE. *Journal of Computational and Graphical Statistics*, 26(2):403–413, 2017.

[3] Francesco Locatello et al. Boosting black box variational inference. In *Advances in Neural Information Processing
Systems*, pages 3401–3411, 2018.


[Meta-Review · NeurIPS 2020]

The reviewers are all aligned in thier positive outlook on this work. I agree with the authors that the proposed new title of this manuscript would increase interest and contextualize the manuscript better. I congratulate the authors on this interesting line of research.